# Informed Consent in the Newly Established Biobank

**DOI:** 10.3390/ijerph16203943

**Published:** 2019-10-16

**Authors:** Judita Kinkorová, Ondřej Topolčan, Radek Kučera

**Affiliations:** 1Laboratory of Immunoanalysis, University Hospital in Pilsen, E. Beneše 13, 30599 Pilsen, Czech Republic; topolcan@fnplzen.cz (O.T.); kucerar@fnplzen.cz (R.K.); 2Department of Medical Chemistry and Biochemistry, Faculty of Medicine in Pilsen, Charles University, Karlovarská 48, 30166 Pilsen, Czech Republic

**Keywords:** Biobank, informed consent, ELSI, personalized medicine

## Abstract

Informed consent is an important document for every existing biobank or repository to obtain, store and use human biospecimens and associated data and information for current and future research. Biobanks have undergone great progress worldwide during the last three decades and have become an integral part of personalized medicine and health care systems and due to new scientific and social questions to be solved. Ethical, legal and social issues (ELSI) guarantee safe sample and data management, and informed consent is one of the key ELSI documents. The form and the consent given in biobank informed consent forms differs between biobank-related institutions, national biobanks, between EU states, and to find the optimal informed consent requires one to fulfil national and international laws and regulations. The Biobank in Pilsen, Czech Republic was officially opened on April the 20th 2017 as a hospital-integrated biobank, and the informed consent was one of the essential documents that had to be ready prior the opening. The process of formulating informed consent corresponding with institutional, national, and international rules and laws to share the experience, to present the challenges, and to demonstrate the national dissimilarity are tasks of the article.

## 1. Introduction

Biobanks are considered one of pillars in personalized medicine and an important source for current and future research. Human biobanks are repositories of human biological material, data and information, it means, biobanks are of dual nature. Biobanks are not static; both samples and data are being collected continuously and repeatedly and due to this biobanks must be open to new techniques, technologies and innovative solutions. During past 15 years considerable changes have occurred regarding biological material, as new genetic and genomic technologies have been developed and used. At the same time extensive databases containing large amount of genotypic and phenotypic data have been developed [1]. It follows that special ethical, legal and social issues (ELSI) must be fulfilled to protect person (data subject = the person the data pertains to) [2], biobank donor [3] and his/her personal data. The most important ethical issue document is an informed consent. Informed consent for the storage and uses of human biological material and related data for the purpose of research is signed by a person who is the donor of biological material for the biobank. Informed consent is given for collection, retention, and use of specimens and it is a process that offer donors information sufficient to allow them to make an informed choice about whether to donate specimens and data to the repository and agree, where applicable, to future research use. Consent should only be obtained under circumstances that provide the prospective donor or the donor´s representative enough opportunity to consider whether to donate and minimizes the possibility of coercion or undue influence [4]. Several types of informed consent are currently used worldwide: *specific consent* for specific research, *broad consent* for future research, *partially restricted consent* to the use of biological specimens and related data for in specific immediate research purpose, *multi-layered consent* (= tiered consent) [5,6] that requires several options to be explained to the research subject, *blanket consent*—open ended permission without any limitations [7]. Generally accepted informed concept in many countries in Europe is a modified broad consent, depending on the biobank or institution. Informed consent should protect the donor privacy and human dignity, and respect social and cultural aspects.

The importance of informed consent and its “form for purpose” and institution is described in detail in Salvaterra et al. [8]. The comprehensive study pointed out the main steps in the development of informed consent forms, the international institutions involved in the process as Council of Europe (COE), World Health Organization (WHO), international laws, guidelines and regulations on biobank-based research and consent requirements, and types of consent. 

Big international biobank bodies like the biggest European biobank infrastructure Biobanking and BioMolecular resources Research Infrastructure—European Research Infrastructure Consortium (BBMRI-ERIC), and International Society for Biological and Environmental Repositories (ISBER) offer support and help in preparation of informed consent documentation [4].

### Biobank in Pilsen

The biobank in Pilsen as a hospital-integrated biobank has been successively built based on previous samples obtained from patients at the Department of Immunochemistry, University Hospital in Pilsen. Since 2000 samples have been systematically collected and stored long-term for routine diagnostics and research. The main diagnoses of interest were breast cancer, prostate cancer, colorectal cancer and lung cancer. In 2014 the repository of biological material at University Hospital in Pilsen changed its status to “biobank” and became a member of the Czech national BBMRI-ERIC node, the biggest biobank infrastructure in Europe [9].

The Czech national node BBMRI_CZ has been created as a network of individual biobanks where each biobank stores samples obtained from associated healthcare providers [10]. Members of the Czech national node are: Masaryk Memorial Cancer Institute (MMCI) in Brno, coordinator of the national node, and associated partners, three medical faculties of Charles University Prague: Bank of Biological Material 1^st^ Faculty of Medicine Charles University Prague (BBM 1FM CU), Bank of Biological Material of the Faculty of Medicine at Hradec Králové Charles University (FM HK CU), and the Bank of Biological Material of the Faculty of Medicine at Pilsen Charles University (FM CU Pilsen], the last partner is Bank of Biological Material at Palacký University in Olomouc (FM PU Olomouc) [11].

The Biobank at University Hospital in Pilsen was officially opened on the 20th April 2017 as a hospital-integrated biobank. Each biobank–member of the Czech national BBMRI_CZ node has its own arrangement, management and structure and provides patients with its own informed consent.

## 2. Material and Methods

Informed consent was one of the first steps in building the biobank in Pilsen together with other tasks that had to be fulfilled to be a full member of BBMRI-ERIC Czech national node. Currently there is no generally accepted “informed consent” for the biobanks at a national level in the Czech Republic. The formulation of informed consent for the biobank in Pilsen started within the institutional Ethical Committee with the discussion which of currently used informed consents is the best possible for the Biobank at University Hospital in Pilsen? Several sets of ethic guidance define the required criteria for consent [12]. Some international guidelines also specify the criteria for informed consent in biobanks [13]. Until now no unified specific guidance for biobank informed consent procedures can be employed [14]. Different research groups in Europe and worldwide propose a unified consent model or possible content for a consent form in biobank research, e.g. for whole genome sequencing studies [15], or genetic research [16].

Based on the general European approaches and based on national requirements, and University Hospital in Pilsen internal rules, the management of the biobank, in close cooperation with the Ethical Committee, decided to take in consideration three approaches: the use and adaptation of informed consent currently used in University Hospital in Pilsen, the use one of the informed consents of the national node partners (BBMRI_CZ), and finally to draw inspiration from the international experience, primarily the BBMRI-ERIC ELSI documents, and ISBER ELSI documents.

The University Hospital in Pilsen, like every other hospital in the Czech Republic, has incorporated informed consent into the whole health care system, regularly revised by the hospital Ethical Committee and patients and published on its web pages in the Czech language [17]. Informed consent as a part of “patient rules” is published on the web pages of the Czech Ministry of Health. This approach reflected hospital and national requirements for the informed consent form.

The biobank in Pilsen as a member of the BBMRI-ERIC Czech national node that takes the lead to use the ELSI services, that supports the biobanking community by facilitating compliance with regulatory requirements and best practices. This approach reflected the requirements of generally accepted European requirements for informed consent.

ISBER is a leading international forum for promoting high standards and innovation in biobanking. For the informed consent preparation, the “Section L: Legal and Ethical Issues for biospecimens, L2: Collection of Human Specimens, L2.2. Informed consent”, p. 78 of Fourth Edition of Best Practices: Recommendations for Repositories was used [4]. This approach completed the basic national, European and international informed consent requirements.

## 3. Results

### 3.1. University Hospital Informed Consents

At University Hospital in Pilsen 614 active informed consents are available for 614 medical acts, and the only “general informed consent” is the consent “with hospitalization ”, all others are related to a single medical intervention. These informed consents were not acceptable for informed consent for the biobank, because they are consents with examination, vaccination, surgery, therapy, which means only for medical interventions that are realized in the hospital or adjacent facilities. Informed consent for biobanks differs in the consent to donate, preserve and provide patients’ samples for current and future research at national or international level together with well-protected patient´s information. Before the first draft of the informed consent was formulated and afterwards compared with informed consents of the Czech node biobank partners, the University Hospital Ethical Committee in Pilsen approved it together with the biobank management members.

### 3.2. Czech National Node Partners Informed Consents

The second step was to collect and compare informed consents from the national node biobank partners. Every biobank of biological material uses different informed consents. Informed consents differ in the title, MMCI does not use the term consent but rather “record of consent…” BBM 1FM CU, and FM CU Pilsen (draft version) use “Consent…“ and the informed consent of FM PU Olomouc use a document which is titled with the Czech Health Agency abbreviation (AZV) and the number of the document. The only institution that has in the title “Informed consent…“ is FM HK CU. All partners differ significantly in the content of the informed consent: MMCI asks patients to consent with preservation and use of unutilized residua of a patient´s body obtained during regular diagnostics or medical treatment for medical research. On the other hand, BBM 1FM CU specifies in the consent the biological material: venous blood (max. 16 mL) and tissue for genetic examination and for research of human diseases. Very specific informed consent is used in FM HK CU; consent with cryopreservation of tumor tissue. The cryopreservation of patient´s tissues will help in the choice of optimal treatment, and for the possible research of new diagnostic methods in molecular biology and thus enhance the quality of the treatment. FM CU Pilsen (draft) asks patients for consent for the collection and preservation of biological material (meaning venous blood, serum and plasma) for clinical research purposes. FM PU Olomouc has a simple consent asking patients to agree with storage of unutilized residua of the patient’s body obtained during regular diagnostics and treatment for research purposes. FM PU Olomouc collects wide range of biological material for a wide range of diagnoses (see Table 1). Table 1 lists the differences between national biobanks as regards the materials collected and diagnoses to better understand the differences between informed consents and the difficulty to harmonize ICs, even at a national level.

Every biobank has currently in its informed consent a possibility to withdrawal consent at any time with no further consequences. Patient’s data protection and information protection are also an issue, where biobanks are not unified and follow their own institutional regulations. Generally, patients’ data are pseudonymized and only accessible by instructed and trained medical staff.

Biobanks in the Czech national node can ask for help and support as regards ESLI a Chief Legal Officer BBMRI-ERIC, Common Service ELSI Expert at Masaryk Memorial Cancer Institute (MMCI) in Brno.

### 3.3. Informed Consent and Patient Requirements

As informed consent is a document that must assure patients that their samples and data will be protected against misuse, will be safely stored and will be used for research, the formulation requires specific attention. Patients must be well informed about benefits and/or risks, about the research goals, must feel the respect to him/her, and to be sure about transparency of the whole process form taking the sample to its use. Information in informed consent must meet the institutional and national ethical, legal and social requirements and recommendations. Valuable information about how to formulate the informed consent were taken from the Beskow et al. study [18].

### 3.4. Patient Involvement

During the preparation phase (about a year) the patients at University Hospital in Pilsen were informed about the aim of the newly formed hospital-integrated biobank and asked about their opinion in a short questionnaire. Patients voluntarily reflected their opinions (about 60 questionnaires) and their answers regarding their willingness to donate their samples for the scientific research were taken in consideration: the length should not exceed one page (format A4), should be simple and easy to read and easy to understand, with explicitly defined purpose, patient’s wish to sign informed consent only once if possible, to guarantee personal data protection, and to have the right to withdraw anytime without any further consequences. The main requirements was to have a right, possibility and anytime to discuss their questions, doubts, and any lack of clarity with an authorized person in the laboratory, and/or biobank.

### 3.5. BBMRI-ERIC ELSI Documents

The third step in creating the informed consent for Biobank in Pilsen was to check informed consents from biobanks in Europe, and BBMRI-ERIC ELSI documents, especially The EU General Data Protection Regulation (GDPR) adopted by the EU member states on 25 May 2018 [19]. The most often used informed consents in European biobanks are broad consents [3]. Grady et al. [7] define broad consent as consent for an unspecified range of future research subject to a few consent and/or process restrictions. Broad consent is less specific than consent for each use, but narrower then open-ended permission without any limitations, i.e. “blanket” consent.

Tiered-layered-staged consent is a model proposed by Bunnik et al. [5,6] for personal genome testing, as an attempt to provide information which is as complete as possible while remaining understandable.

Dynamic consent was pointed out by Kaye et al. [20] as an example of how information technologies (IT) that can be used safely to satisfy the legal and regulatory requirements for research consent in the twenty first century are promising the next step in the evolution of informed consents, but because of many obstacles, as e.g. modern web-based technologies available both for the medical staff and for patients, implementation is time- and financially-demanding, so it was not able to take it into consideration.

A good example for formulation of informed consent in the Biobank University Hospital Pilsen was the “Template for informed consent concerning the donation, storage, and utilization of biological material as well as collecting, processing, and usage of (related) data in biobanks”, version 2.0 recommended by the Permanent Working Party of the German Medical Ethics Committees approved by the General Assembly on 10 June 2016. Informed consent for Biobank in University Hospital in Pilsen was formulated based on the three subsequent steps described above and published prior the official opening of the biobank.

## 4. Discussion

Informed consent is an ethical and legal requirement for research involving human participants in medical research [21], and the dynamic development of biobanks requires them to obtain an informed consent (it refers to the issue for future scientific researchers, the access of scientists from other institutions) to ensure the privacy, the right not to know, and the possibility of the commercial usage of the samples [22].

Many newly established biobanks all over the world have described their approaches how to solve ethical, legal and social issues, and informed consent was one of the most difficult tasks [21,22,23,24,25,26].

Currently the informed consent used in Biobank University Hospital in Pilsen was formulated based on process of subsequent steps from institutional and national to European and international concepts as a compromise between all the approaches and models mentioned above. The closest form is the broad consent. During the process comprehensive approach was chosen. First the requirements of the University Hospital Ethical Committee were taken in consideration, then national legislation, which means laws, regulations and recommendations and demands, were reflected even though in the Czech Republic there is no Czech national law on biobanks yet. Also, the informed consents of Czech partners in National BBMRI_CZ node were carefully studied and were taken for inspiration.

We had to take in consideration the LIMS system in the University Hospital in Pilsen, also the time for informed consent preparation, available and appropriate personnel, financing and others.

As a member of the biggest European biobanking research infrastructure BBMRI-ERIC, we turned to the ELSI department that provides useful and efficient services and tools for biobanks to answer ELSI matters to fulfil the informed consent requirements at European level.

ISBER publishes periodically “Best Practices: Recommendations for Repositories” a guidance document that reflects the collective experience of its members and has received broad input from other repository professionals. Basic information about informed consent, its types and requirements and rules were published in the Fourth Edition of Best Practices [4]. ISBER publishes regularly its own journal *Biopreservation and Biobanking* and ethical issues are of high priority [1].

During the formulation of informed consent for Biobank in Pilsen the wide discussion about forms, extent and content was followed both at literature and within the scientific community of the International Society for Biological and Environmental Repositories (ISBER), and the scientific discussion demonstrates that informed consent and other ethical, legal, and social issues are very dynamic topic reflecting rapid progress in current biobanking practices.

The current version of informed consent for the biobank in Pilsen will undergo a testing process and will be evaluated and actualized based on the experience and new requirements of medical doctors, researchers and patients, as well as national and international demands.

## Figures and Tables

**Table 1 ijerph-16-03943-t001:** Biological material collected in biobanks.

Institution Name	Biological Material Collected	Diagnoses
Masaryk Memorial Cancer Institute (MMCI) Brno	liquid biological material, frozen tissue, formalin-fixed paraffin-embedded (FFPE) tissue, bone marrow, genetic material	oncological
1st Faculty of Medicine Charles University and General University Hospital in Prague (BBM 1FM CU)	liquid biological material, frozen tissue, bone marrow, genetic material	oncological, rare
Faculty of Medicine at Hradec Králové Charles University (FM HK CU)	liquid biological, frozen tissue material, FFPE tissue	cardiovascular, oncological, rare, metabolic
Faculty of Medicine at Pilsen Charles University (FM CU Pilsen)	liquid biological material, FFPE tissue, genetic material	oncological neurodegenerative cardiovascular
Palacký University in Olomouc (FM PU Olomouc)	liquid biological material, frozen tissue, FFPE tissue bone marrow, genetic material, cell lines	oncological neurodegenerative cardiovascular, rare

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
