# Peer review of "Informed Consent in the Newly Established Biobank"

_ijerph, 2019, doi:10.3390/ijerph16203943_

Round 1

Reviewer 1 Report

I still sustain my previous opinion about the manuscript I got to review.

Reviewer 2 Report

The revised version significantly improves the presentation of the argument and provides the reader with all necessary background information. The authors were also able to substantiate their findings thus improving the soundness of their claims. 

My only (minor) criticism is that the authors could be a little bit more "bold" in the discussion. Their careful analysis of one particular case study produces some very interesting results. Perhaps they could add one or two sentences stressing what others could learn form the Czech example? 

This manuscript is a resubmission of an earlier submission. The following is a list of the peer review reports and author responses from that submission.

Round 1

Reviewer 1 Report

Kinkorova J et al. wrote a manuscript about a biobank project of Czech Republic. I am sorry that I could not understand the importance of publishing this manuscript with following reasons.

First paragraph of the introduction section may be lengthy. I think that general explanations of purpose and methodologies of biobank wound not be needed. They could be shrunk. I understand that this manuscript is written to clarify the importance of informed consent. Is it accurate understanding? If it is, the manuscript need to explain why the new informed consent would contribute to the field of medical science, or why the manuscript need to be widely read for the science. Table: I was unable to understand new findings and what to tell to the other researchers of this manuscript. 

Reviewer 2 Report

The authors describe in detail the ethical issues which are very important part of biobanking human tissues specimen for the research in the future. Collected human tissues together with the information about the clinical course of disease and demographic data about the donors are of great value for the progress in medical sciences. The authors describe the bioethical options worked out by different biobanks together with European biobanking organization BBMRI-ERIC since the Czech republic (as other European countries) doesn’t have the legal acts on biobanks and bioethics.

The manuscript contains important information for researchers who would like to establish and maintain the repositories of human samples stored for the future research. The ethical regulations are important for publishing the results of research performed on the biobank samples collections.

Reviewer 3 Report

The article focuses on an interesting topic: the drafting/selection of "informed consent" (IC) for potential donors to a biobank. the authors are right to stress the institutional frameworks and "best practices".

However I´d like to encourage them to put less emphasize on the "description" and more on the actual analysis. For instance, the "Material and Method" section should be expanded. They could mention the criteria for assessing the "three approaches" and explain the process of selecting IC´s in greater detail, to help readers understanding the "results" section better.  

Moreover, the "results" section should be elaborated as well. The last sentence in section 3.1 remains unclear: Why werent these IC´s deemed unacceptable. If possible the authors should also explain the final decision making process in greater detail. Under 3.4 they simply remark that "IC (...) was formulated based on above described documents..." But why these documents? Was it a consensual process or a compromise? Who made the final decision and who was drafting the IC? 

Explaining the circumstances better would increase the relevance of the case Biobank in Pilsen.